# TP53 mutations emerge with HDM2 inhibitor SAR405838 treatment in de-differentiated liposarcoma

Joonil Jung[1], Joon Sang Lee[1], Mark A. Dickson[2], Gary K. Schwartz[3], Axel Le Cesne[4], Andrea Varga[4], Rastilav Bahleda[4], Andrew J. Wagner[5], Edwin Choy[6], Maja J. de Jonge[7], Madelyn Light[1], Steve Rowley[1], Sandrine Macé[8] & James Watters[1]

In tumours that harbour wild-type p53, p53 protein function is frequently disabled by the mouse double minute 2 protein (MDM2, or HDM2 in humans). Multiple HDM2 antagonists are currently in clinical development. Preclinical data indicate that TP53 mutations are a possible mechanism of acquired resistance to HDM2 inhibition; however, this resistance mechanism has not been reported in patients. Utilizing liquid biopsies, here we demonstrate that TP53 mutations appear in circulating cell-free DNA obtained from patients with de-differentiated liposarcoma being treated with an inhibitor of the HDM2–p53 interaction (SAR405838). TP53 mutation burden increases over time and correlates with change in tumour size, likely representing selection of TP53 mutant clones resistant to HDM2 inhibition. These results provide the first clinical demonstration of the emergence of TP53 mutations in response to an HDM2 antagonist and have significant implications for the clinical development of this class of molecules.

[1] Sanofi Oncology, Cambridge, Massachusetts 02139, USA. [2] Department of Medicine, Memorial Sloan Kettering Cancer Center, New York, New York 10065, USA. [3] Columbia University Medical Center, New York, New York 10032, USA. [4] Department of Cancer Medicine, Gustave Roussy, Villejuif 94800, France. [5] Center for Sarcoma and Bone Oncology, Department of Medical Oncology, Dana–Farber Cancer Institute, Boston, Massachusetts 02215, USA. [6] Department of Medical Oncology, Massachusetts General Hospital, Boston, Massachusetts 02114, USA. [7] Department of Medical Oncology, Erasmus MC Cancer Institute, Rotterdam 3015 CE, The Netherlands. [8] Sanofi Oncology, Vitry-sur-Seine 94400, France. Correspondence and requests for materials should be addressed to J.J. (email: joonil.jung@sanofi.com) or to J.W. (email: james.watters@gmail.com).

nactivation of p53 function is an almost universal feature of human cancer cells. While loss of tumour suppressive function of p53 is often due to somatic mutations, approximately half of all tumours still harbour wild-type p53 (refs 1,2). In p53 wild-type tumours, biological function is frequently disabled by the mouse double minute 2 protein (MDM2, or HDM2 in humans)[3]. Therefore, disruption of the interaction between p53 and HDM2 with small molecules, and subsequent reactivation of p53, is an attractive treatment strategy. Preclinical studies have demonstrated that *TP53* mutations are a possible mechanism of acquired resistance to HDM2 antagonists in osteosarcoma, rhabdomyosarcoma, neuroblastoma, melanoma and lymphoid tumour models[4–7]. Multiple HDM2 antagonists are currently in clinical development; however, *TP53* mutation as a mechanism of resistance has not been reported in patients.

SAR405838 is an oral spirooxindole derivative inhibitor of the HDM2–p53 interaction (Fig. 1a). SAR405838 is undergoing evaluation in a phase 1 trial in which the maximum tolerated dose (MTD) was established as 300 mg once daily (NCT01636479); 21 patients with de-differentiated liposarcoma (DDLPS) were enrolled in an MTD expansion cohort to assess efficacy in patients whose tumours exhibited genomic amplification of *MDM2*. Pharmacokinetics (PK), pharmacodynamics and efficacy data have been reported elsewhere[8]. The clinical benefit of SAR405838 treatment was modest, with multiple patients experiencing disease progression within 12 weeks and no objective responses observed[8] (manuscript in preparation). Therefore, we sought to investigate potential mechanisms of resistance to SAR405838. We found that *TP53* mutations appear in circulating cell-free DNA (cfDNA) obtained from patients with DDLPS being treated with SAR405838. *TP53* mutation burden increases over time and correlates with change in tumour size, likely representing selection of *TP53* mutant clones resistant to HDM2 inhibition. These results provide the first clinical demonstration of the emergence of *TP53* mutations in response to an HDM2 antagonist and have significant implications for the clinical development of this class of molecules.

## Results

**Tumour and liquid biopsies used for mutation analysis.** Baseline tumour biopsies were obtained from 20 patients, 17 of which yielded sufficient DNA for genetic analysis. *MDM2* amplification was detected in 15 patients (89%) and no somatic mutations were identified in *TP53* (Table 1), confirming the high prevalence of the target genetic profile in this indication.

Given the limitations of serial tumour biopsies[9], we used 'liquid biopsies' to assess the emergence of *TP53* mutations in patients being treated with SAR405838. A number of methods, including BEAMing or digital PCR, allow highly sensitive detection of mutations but typically require prior knowledge of the specific mutation(s) of interest[9,10]. Thousands of mutations in *TP53* have been reported in public databases, with ~80% of these mutations occurring in the DNA-binding domain (http://p53.iarc.fr/). We developed a custom next-generation sequencing assay covering all coding exons and untranslated regions of *TP53* to assess mutation acquisition in an unbiased manner.

**Variant allele frequency threshold for *TP53* variants.** To determine the variant allele frequency (VAF) threshold for declaring a mutation, we first sequenced cfDNA from 10 healthy volunteers. Although multiple low-frequency variants in *TP53* were called, most variants exhibited strong strand bias suggestive of being sequencing artefacts. In addition, no variant was observed above 0.5% VAF (Fig. 1b). To assess the proportion of mutations in cfDNA that are also detected in matched tumours,

we sequenced 60 matched tumour and plasma pairs from patients with colorectal or non-small cell lung cancer. We observed that VAF >1% exhibited significantly reduced strand bias (Fig. 1c). Using a VAF cutoff of 1%, 18 *TP53* variants were called in cfDNA, of which 13 were also identified in matching tumour samples (72%; Fig. 1d). Conversely, only 8/208 variants at VAF <1% in cfDNA were identified in matching tumour samples (3.8%). Therefore, we set a conservative VAF threshold of 1% for calling *TP53* variants in cfDNA.

**TP53 mutations emerge during treatment with SAR405838.** Twenty-six *TP53* mutations were identified in cfDNA samples from patients undergoing treatment with SAR405838 in the MTD expansion cohort (Fig. 2a). Consistent with several preclinical studies[5,6], all were missense mutations in the DNA binding domain, resulting in the alteration of 14 different amino acids (Fig. 2b). All *TP53* mutations have been previously reported in COSMIC (http://cancer.sanger.ac.uk/cancergenome/projects/cosmic/) and are predicted to have a deleterious impact on protein function[11], resulting in inability of p53 to bind target sequences and transactivate target genes.

Although 26 mutations in *TP53* were identified, these mutations clustered in only five patients (Fig. 2c). No mutations were observed in the baseline plasma samples, in agreement with the *TP53* wild-type status observed in baseline tumour samples, indicating that *TP53* mutations emerge after the initiation of SAR405838 treatment (posterior predictive probability, $P = 0.0017$). Multiple *TP53* mutations emerged within individual patients, including independent variants altering the same p53 residue, and mutations appeared as early as 6 weeks after treatment initiation. There was a significant increase in total *TP53* mutation burden with time, with the exception of patient 10, whose allele burden peaked at 24 weeks.

Inspection of sequencing alignments indicated that multiple *TP53* mutations present within individual patients likely represent independent clones (Supplementary Fig. 1). Of five patients with at least one sample collected on or after 12 weeks of treatment (Supplementary Table 1), four showed the emergence of *TP53* mutations at VAF >1% (Fig. 2d). These results demonstrate that patients being treated with SAR405838 acquired inactivating *TP53* mutations that increased over time, likely representing the outgrowth of resistant clones in response to selective pressure applied by activation of p53.

**Association of clinical variables with *TP53* mutations.** An analysis of clinical variables revealed that increase in tumour size by computed tomography significantly correlated with increased *TP53* mutation burden (Fig. 2e, Supplementary Fig. 2). Since four out of five patients with the emergence of *TP53* mutations in cfDNA had stable disease at 12 weeks (PFS-12w, Supplementary Table 1), we tested if PK might play a role in the development of *TP53* mutations. However, no significant difference in initial $C_{max}$ was observed between patients with or without PFS-12w (mean ± s.d.: 1,497 ± 857 versus 1,606 ± 633 ng ml$^{-1}$, Cycle 1, Day 1, number of patients = 20, Supplementary Fig. 3). We compared tumour volume at baseline between patients that did or did not go on to acquire *TP53* mutations but found no significant difference (668.4 ± 648.6 versus 422.18 ± 646.2 cm$^3$, Supplementary Table 1). Similarly, there was no significant difference in tumour volume among the samples with or without *TP53* mutations in the cfDNA (549.8 ± 649.1 versus 466.8 ± 629.0 cm$^3$, Supplementary Table 1). Our hypothesis is that *TP53* mutations emerge with a longer time on SAR405838 treatment. Therefore, it is possible that the patients treated for <12 weeks were not exposed to SAR405838 long enough to develop *TP53* mutations.

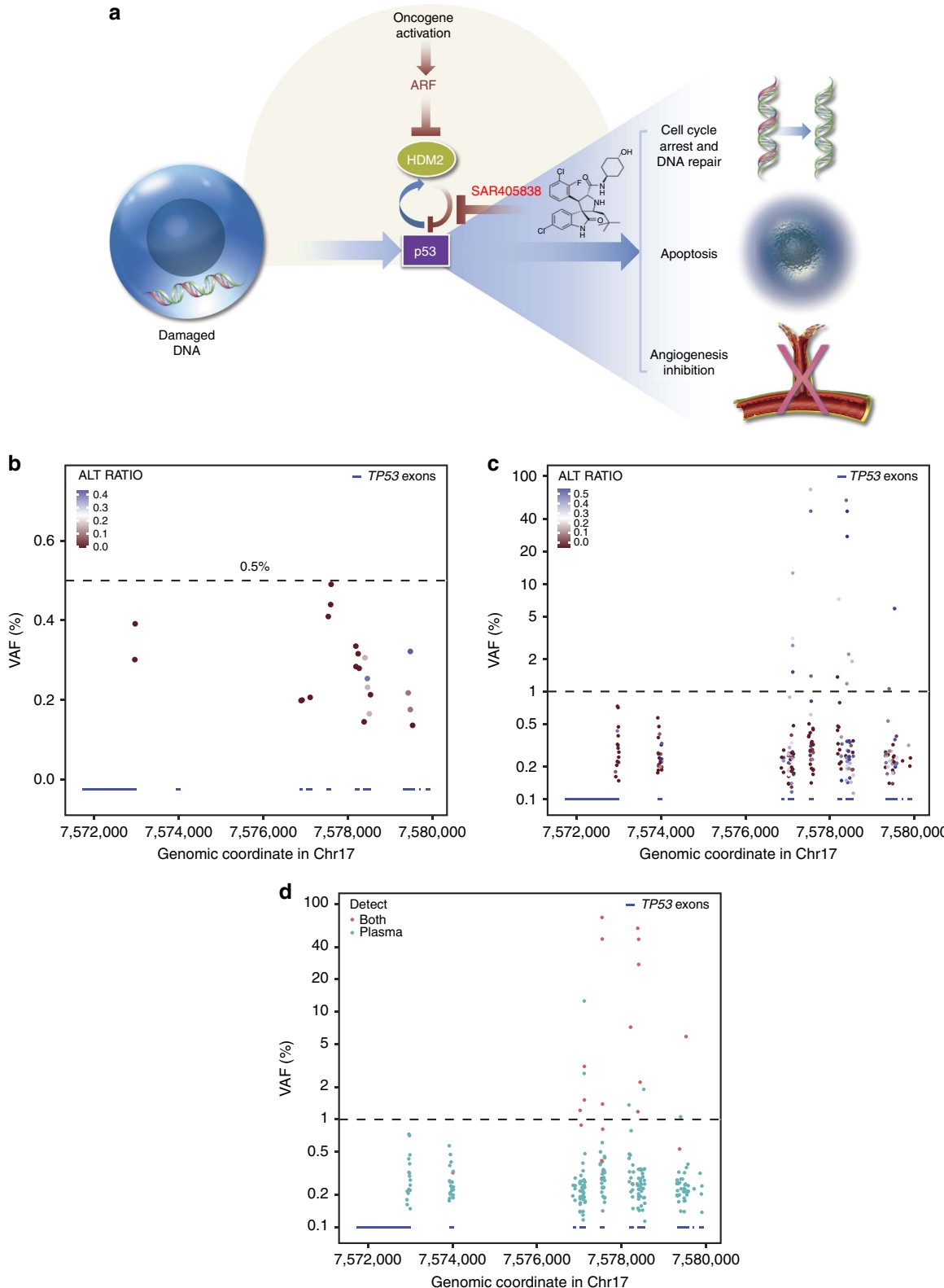

**Figure 1 | SAR405838 mode of action and determination of VAF threshold for *TP53* mutations.** (**a**) SAR405838 inhibits the interaction between HDM2 and p53, resulting in activation of p53 function. (**b**) *TP53* mutation VAF in cfDNA samples from normal healthy volunteers (*n* = 10). The *X* axis shows the genomic location of *TP53* exons and UTRs (blue bars). Each dot indicates the presence of one *TP53* variant. ALT_RATIO, a measure of strand bias, is defined as the proportion of reads in the less-abundant read direction at a base where a variant is detected. A ratio of 0.5 = no strand bias (blue). The dotted line indicates a VAF of 0.5%. (**c**) *TP53* mutation VAF in cfDNA samples from 60 matched CRC and NSCLC tumour/plasma pairs. Each dot indicates the presence of one *TP53* variant. The dotted line indicates a VAF of 1%. (**d**) *TP53* mutation tumour concordance in cfDNA samples from 60 matched CRC and NSCLC tumour/plasma pairs. Each dot indicates the presence of one *TP53* variant in cfDNA. Red dots indicate variants that were also present in the matched tumour sample. The dotted line indicates a VAF of 1%. cfDNA, cell-free DNA; CRC, colorectal cancer; NSCLC, non-small cell lung cancer; VAF, variant allele frequency.

**Table 1 | Baseline tumour *MDM2* and *TP53* genetic status and plasma collection matrix in patients with DDLPS treated with SAR405838.**

| Patient ID | Tumor | | Plasma | | | | | | |
| --- | --- | --- | --- | --- | --- | --- | --- | --- | --- |
| | MDM2 amplification status | p53 mutation status | B/L | 6w | 12w | 18w | 24w | 30w | 36w |
| 1 | Amplified | WT | | | | | | | |
| 2 | N/D | N/D | | | | | | | |
| 3 | Gain (3–5 copies) | WT | ▓ | | | | | | |
| 4 | Amplified | WT | ▓ | | | | | | |
| 5 | Amplified | WT | | | | ▓ | | | |
| 6 | | WT | | ▓ | | | | | |
| 7 | Amplified | WT | ▓ | | | | | | |
| 8 | Amplified | WT | ▓ | | | | | | |
| 9 | Amplified | WT | | ▓ | | | | | |
| 10 | Amplified | WT | | ▓ | ▓ | ▓ | ▓ | ▓ | ▓ |
| 11 | Amplified | WT | | | | | | | |
| 12 | Amplified | WT | | | ▓ | ▓ | ▓ | ▓ | ▓ |
| 13 | N/D | N/D | | | | | | | |
| 14 | Amplified | N/D | ▓ | ▓ | | | | | |
| 15 | Amplified | WT | ▓ | ▓ | ▓ | ▓ | | | |
| 16 | Amplified | WT | ▓ | ▓ | | | | | |
| 17 | Amplified | WT | ▓ | | | | | | |
| 18 | Amplified | WT | | ▓ | | | | | |
| 19 | Amplified | WT | ▓ | | | | | | |
| 20 | N/D | N/D | ▓ | | | | | | |

B/L, baseline; DDLPS, de-differentiated liposarcoma; N/D, not determined; WT, wild type.
Grey boxes indicate time points for which plasma samples were collected.

**Origins of *TP53* mutant DNA in cfDNA**. On inspection of tumour sequencing data, there was no clear evidence that emergent cfDNA mutations were present at low levels in the tumour at baseline. This could be because the tumour was not sequenced with an adequate sensitivity to detect rare clones, the tumour biopsy was taken from a region that did not harbour specific mutations, or because these mutations were not present in the baseline tumour. As such, we cannot exclude the possibility that emergent *TP53* mutations represent *de novo* mutations, as described in preclinical studies[5,6].

Recently it was proposed that haematopoietic stem/progenitor cells (HSPCs) in elderly people accumulate mutations in *TP53* and chemotherapy might confer a selective clonal advantage to *TP53* mutant clones[12]. Since the majority of DDLPS patients in our study have received multiple rounds of chemotherapy prior to enrollment (Supplementary Table 1), it is possible that SAR405838 could have conferred a selective clonal advantage to pre-existing *TP53* mutant HSPC clones, which in turn contributed to the emergence and increase in *TP53* mutant alleles in cfDNA. In our study, we evaluated *TP53* mutation status at baseline from 15 patients with prior chemotherapy, either by tumour DNA sequencing or cfDNA sequencing and all were *TP53* wild type (Fig. 1b). Although we cannot rule out the possible existence of minor subclones below the detection limit of our sequencing assays, and subsequent clonal outgrowth of such minor *TP53* mutant HSPC clones during SAR405838 treatment, if *TP53* mutant alleles in cfDNA had indeed originated primarily from *TP53* mutant HSPCs, it is unlikely we would have observed

the correlation between *TP53* mutation burden and tumour growth (Fig. 2e). In addition, an increase in transversion frequency to ~50% has been reported in patients after chemotherapy[13]. While the sample size is small, the frequency of transversion events in *TP53* in our study is only 20% (3/15, posterior predictive probability $P = 0.01757812$ under the assumption that transversion frequency = 50%; Supplementary Data 1). We believe our data argue that the *TP53* mutant alleles observed in five patients originated from tumour tissue. However, comparisons between cfDNA and post-treatment tumour biopsies or peripheral blood mononuclear cells will be needed to definitively test the contribution of mutant HSPC clones to the emergence of *TP53* mutants in response to HDM2 inhibition.

## Discussion

Our results are consistent with several preclinical studies[5,6,14], and provide the first *in vivo* evidence in support of the hypothesis that acquired mutations in *TP53* contribute to HDM2 inhibitor resistance in the clinic. While the correlation between *TP53* mutation burden and tumour growth in this study does not prove causality, our results could have significant implications for the clinical development of SAR405838 and other HDM2 antagonists. We hypothesize that there is no selective pressure for tumours to remain *TP53* wild type, enabling the rapid emergence of resistance mutations. Given the rapidity with which *TP53* mutations emerged, we hypothesize that HDM2 antagonists will likely require development in combination with other agents. Various combination partners, including a mutant p53 activating

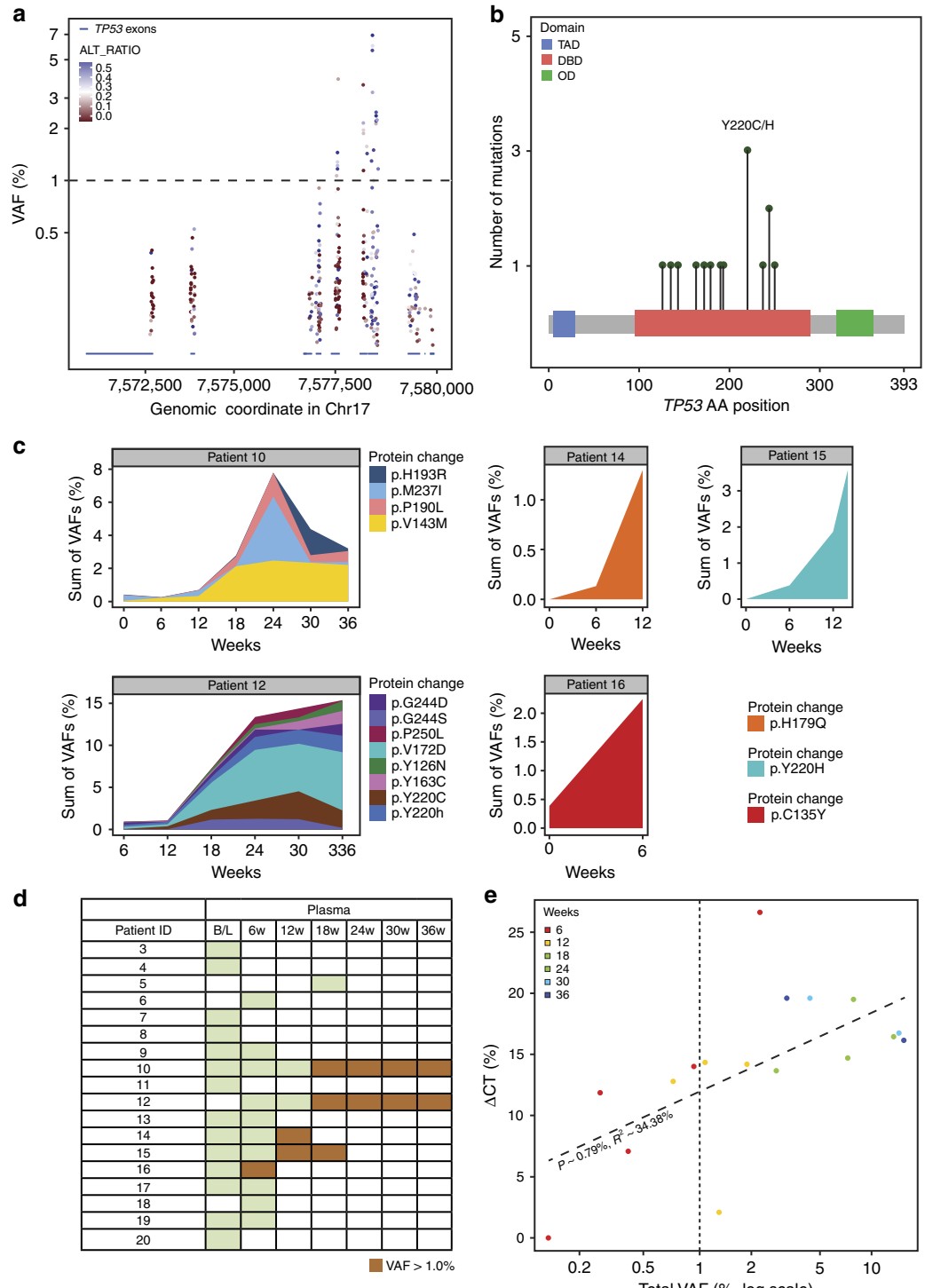

**Figure 2 | *TP53* mutations in patients with DDLPS treated with the HDM2 antagonist SAR405838.** (**a**) *TP53* mutation VAF in cfDNA samples from patients with DDLPS in the SAR405838 MTD expansion cohort. The *X* axis shows the genomic location of *TP53* exons and UTRs (blue bars). Each dot indicates the presence of one TP53 variant. Dotted line indicates VAF of 1.0%. (**b**) All identified mutations in *TP53* were located in the DNA binding domain. The *Y* axis is the number of mutations observed at each position; the *X* axis represents the amino-acid sequence of *TP53*. Each green dot represents a missense mutation. The most frequently altered amino-acid position (Y220) is indicated. (**c**) *TP53* mutations at VAF >1.0% were clustered in five patients. The *Y* axis shows the sum of *TP53* mutation VAF (allele burden) and the *X* axis shows weeks of treatment. Each mutation is represented by one colour. (**d**) Patient *TP53* mutation status. Green indicates time points for which cfDNA samples were collected and sequenced, but no *TP53* mutations were identified at VAF >1.0%. Orange indicates time points for which *TP53* mutations were identified at VAF >1.0%. (**e**) Correlation between percent change in tumour size by CT (the *Y* axis) versus total *TP53* mutation VAF (mutation burden; the *X* axis, log scale) across all patients for which at least one *TP53* mutation was identified at VAF >1%. Each dot represents one time point for one patient and dots are coloured according to the time point. The vertical dotted line indicates *TP53* mutation VAF = 1.0%. B/L, baseline; cfDNA, cell-free DNA; CT, computed tomography; DBD, DNA binding domain; DDLPS, de-differentiated liposarcoma; MTD, maximum tolerated dose; OD, oligomerization domain; TAD, transactivation domain; VAF, variant allele frequency.

drug like APR-246, MEK or PI3K inhibitors, or Bcl-2 inhibitors, have been proposed[5,15,16]. Ongoing clinical studies may help determine if combinations can improve the clinical activity of HDM2 antagonists.

## Methods

**Patients and samples.** Patients were part of a phase 1 study testing the safety of SAR405838 (ClinicalTrials.gov Identifier: NCT01636479). Safety, pharmacokinetics, pharmacodynamics and preliminary efficacy for this phase I study have been previously reported[8]. Patients with a clinical diagnosis of DDLPS were enrolled in a MTD expansion cohort (300 mg once daily) to assess the biological activity of SAR405838. SAR405838 was administered orally, and one cycle was defined as 21 days of treatment. Tumour progression was assessed according to Response Evaluation Criteria In Solid Tumours (RECIST) 1.1. RECIST was followed in unidimensional analysis and anatomical assessment of tumour by computed tomography scan at baseline and during treatment period approximately every 6 weeks at participating clinical sites. Pre-treatment surgical or core needle tumour biopsies were obtained from 20 patients participating in the expansion cohort. Tumour biopsies were fixed in formalin and embedded in paraffin (FFPE) blocks. Sample collection for this study was approved by the ethics committees of participating institutions. Sections (5 μm) were stained with haematoxylin and eosin according to standard procedures and examined by an expert pathologist to confirm the diagnosis of DDLPS and to ensure the presence of at least 30% tumour cells. At least five 10 μm scrolls were cut from each FFPE block and placed into sterile 1.5 ml tubes for DNA extraction. DNA was extracted from tumour biopsies using the QIAamp DNA FFPE Tissue kit (Qiagen). Blood (3–4 ml) was collected into EDTA tubes from patients at baseline and before treatment on day 1 of every other cycle for as long as patients participated in the study. Collection of blood samples for plasma isolation and DNA extraction was optional for this study. Informed consent was obtained from all participants.

**Tumour genetic status.** MDM2 amplification in baseline tumour biopsies was assessed by TaqMan quantitative PCR (Applied Biosystems), using RNAseP as the reference gene (MDM2 probe Hs01463512_cn, TaqMan Copy Number Reference Assay RNase P). Relative quantitation was performed using the ΔΔCt method and a normal healthy human donor DNA sample was used as the calibrator. Amplification was defined as having greater than five copies of MDM2 using the mean of triplicate measurements. Tumour mutation profile was assessed using the Ion AmpliSeq Cancer Hotspot Panel v2 (Life Technologies). Tumours were sequenced to a median coverage of at least ×19,000. Mutations were called using MuTect[17], Strelka[18] and SomaticIndelDetector (http://www.broadinstitute.org/cancer/cga/). Oncotator[11] was used to annotate mutation calls. Tumours were declared TP53 wild type if no non-synonymous mutations were called.

**Plasma preparation and cfDNA isolation from plasma.** Plasma was prepared at clinical sites within 15–30 min from blood draw using double centrifugation[19]. Blood samples were processed first by centrifugation at 1,600 (+150)g for 10 min. The supernatant was transferred to a fresh 2 ml tube and was centrifuged again at 3,000 (+150)g for 10 min. The resulting supernatant was transferred into a 3.5 ml polypropylene tube and stored at −80 °C until cfDNA isolation. This process typically yielded ∼1.2 ml of plasma for DNA isolation. For cfDNA isolation, we used either a manual or an automated process. For manual extraction, we used the QIAamp Circulating Nucleic Acid Kit (Qiagen, catalogue # 55114) using the QIAvac 24 Plus (Qiagen, catalogue # 19413) according to the manufacturer's recommended procedures. For automatic cfDNA preparation, we used the QIAsymphony DSP Virus/Pathogen Kit (Qiagen, catalogue # 937055) using the QIAsymphony SP system (Qiagen, catalogue # 9001751). cfDNA from patient samples was extracted using the manual protocol while cfDNA from samples obtained from the commercial provider Proteogenex was extracted using the automated protocol. cfDNA yield was not significantly different between commercially acquired plasma samples and plasma samples obtained from the SAR405838 MTD expansion cohort (Supplementary Fig. 4a). The plasma cfDNA yield from patients participating in the DDLPS expansion cohort did not consistently increase or decrease over time (Supplementary Fig. 4b).

**Genomic DNA purification from commercial FFPE tissues.** FFPE tissues and matched plasma were obtained from Proteogenex (www.proteogenex.com). gDNA was purified from two 10 μm sections of FFPE blocks. Sections were first treated in Qiagen deparaffinization solution (catalogue # 19093) and then processed in the QIAsymphony SP system using the QIAsymphony DSP DNA Mini Kit (Qiagen, catalogue # 61304).

**Targeted sequencing library preparation.** To monitor tumour genetic status using liquid biopsies, we developed a targeted deep sequencing assay for mutation detection based on a hybrid-capture target enrichment strategy. Our custom capture panel covers all coding exons and untranslated regions of TP53 (Supplementary Table 2). We used whole-genome sequencing (WGS) libraries for hybrid-capture of target sequences. WGS libraries were generated using the KAPA Hyper Prep Kits (KAPA Biosystems, catalogue # KK8504). Typically, 5–20 ng of cfDNA was used as input for WGS library preparation. 100 ng of gDNA was used for FFPE tissue samples acquired from Proteogenex. Custom xGEN Lockdown Probes were used for hybrid-capture using the manufacturer's Rapid Protocol Version 2.1 with the following modifications: we used a total of 1 μg of WGS library, typically a pool of three or four libraries, for hybrid-capture and incubated the hybridization reaction at 65 °C (lid 80 °C) for 18–24 h. Typically, we achieved greater than 40% on-target rate and >8,000× mean exonic sequence coverage using our cfDNA sequencing assay (Supplementary Fig. 4c).

**Mutation analysis.** Typically we generated 100 or 150 bp paired-end sequencing reads and trimmed the reads down to 75 bp to retain high-quality base calls. The sequencing reads were mapped to the human reference genome hg19 with the Novoalign3 (http://www.novocraft.com/). After the initial read alignment, indel realignment was performed using ABRA[20] to increase mapping/alignment accuracy[21]. Three mutation callers, MuTect[17], LoFreq[22] and Pindel[23] were used for the detection of single nucleotide variants and indels with the re-aligned Binary sequence Alignment/Map (BAM) files[24]. The resulting mutation calls were annotated by Oncotator[11]. Silent or non-coding mutations were excluded from the analysis along with the known germline TP53 P72R polymorphism. Given the difficulties associated with variant calling when using plasma-derived cfDNA and targeted deep sequencing, we manually inspected variant calls of interest with an integrative genomics viewer[25].

**Statistical tests.** Using the TP53 sequencing results from the nine plasma samples from healthy volunteers as a prior, we calculated posterior predictive probability of observing a TP53 mutation from the patient plasma treated with SAR405838. Let $P$ be the probability that a TP53 mutation is detected from a normal cfDNA sample. Since we did not have any prior information on $P$, a non-informative prior distribution of $P$, a uniform prior would be a good choice for a prior distribution of $P$. Then, given our nine plasma normal sequencing data and uniform prior distribution, the posterior distribution of $P$ was found to be a beta distribution with parameters $b1 = 1$ and $b2 = 10$. We did not find any TP53 mutations from all of 14 baseline plasma samples, whereas TP53 mutations are found from 12 out of 25 treatment plasma samples. The posterior predictive probability of detecting a TP53 mutation from 0 out of 14 baseline plasma samples is 40%. On the contrary, the posterior predictive probability of detecting a TP53 mutation from 12 out of 25 treatment plasma samples is 0.17% (about 231-fold change). This indicates that the latter case is much less likely to occur by chance.

We tested association between time and TP53 mutational burden with a Pearson correlation test, cor.test() in R (ref. 26). This computed a Pearson correlation between the week of observation and the total TP53 VAF observed. A significant $t$-test was performed with N-2 degrees of freedom and an asymptotic confidence interval based on the Fisher Z-transform. Since the 95% confidence interval of the correlation was bounded away from 0, we rejected the null hypothesis of no correlation and accepted the alternative hypothesis that time and mutational burden are related.

We tested association between computed tomography change from baseline and TP53 mutational burden in a linear-log regression model by regressing the computed tomography change from baseline on the base-10 log of the total TP53 VAF, lm() in R (ref. 26). We required that both the overall regression had a statistically significant $F$-test and the slope coefficient had a significant $t$-test. We then checked that a Pearson correlation test was also significant, at approximately the same $P$ value.

The association between PK or tumour volume and the emergence of TP53 mutations was tested by $t$-test (two-sided, s.d. used to indicate error range) and no significant differences were found. Tumour volumes at baseline were compared only when plasma was tested at 6 weeks or later time points and, therefore, baseline tumour volume data for patients #3 and #4 were excluded from the analysis. Tumour volume information for patient #20 was not available and was also excluded from the analyses.

To test the significance of the low transversion mutation rate in our TP53 cfDNA sequencing data compared with the recent publication[13], we performed a binomial test with the assumption that transversion frequency = 50%. The resulting $P$ value is 1.76%.

**Data availability.** All relevant data are available from the authors on request and/or are included with the manuscript (as figure source data or Supplementary Information Files).

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

## Acknowledgements

We acknowledge Barbara Kalinowska for tumour specimen processing and sectioning, and Rafaelle Baffa for pathological review of tumour specimens. We acknowledge Colette Dib, Wilson Dos-Santos-Bele, Sandrine Valence, Edouard Turlotte and Stephane Soubigou for *TP53* and *MDM2* status determination in baseline tumour biopsies. We acknowledge Karl Hsu, Wei Zheng and Koruth Thomas from the clinical study team for providing clinical data for the preparation of this manuscript. This study was funded by Sanofi. We received editorial support from Simone Blagg of MediTech Media, funded by Sanofi.

## Author contributions

All authors contributed to the conception or design of the work, or the acquisition, analysis or interpretation of data for the work. J.J. and J.W. drafted the article. J.S.L., M.L., S.R., S.M., A.J.W., M.A.D., G.K.S., A.L.C., A.V., R.B., E.C. and M.J.d.J. revised the manuscript critically for important intellectual content.

## Additional information

**Competing financial interests:** J.J., J.S.L., M.L., S.R., S.M. and J.W. were Sanofi employees at the time of manuscript writing. A.J.W.: research funding from Sanofi, Novartis, Roche and Amgen, and consulting for Daiichi-Sankyo. M.A.D., G.K.S., A.L.C., A.V., R.B., E.C. and M.J.d.J. have no conflicts of interest.

