## [Peer Review File · Nature Communications]

Reviewers' comments:

Reviewer #2 (Remarks to the Author):

This interesting report describes the emergence of mutations in cfDNA in patients with liposarcoma with HDM2 amplification receiving HDM2 inhibitor SAR405838. The results of the clinical trial in terms of tumor control and side effects are not described, and will be described in a subsequent report. This is an important limitation of the study. Indeed, none of the patients have truly responded to the treatment, and therefore it remains speculative to mention primary or secondary resistance in tumors which are maybe only slowly progressive. An other limitation of the study, pointed out by the authors is that rebiopsies were not performed. How can we be sure that mutations do not arise from other tissues? The correlation with tumor progression is only a correlative not demonstrative.

The method used to assess the presence/emergence of mutated p53 are appropriate. Most mutations are deleterious according to the public databases. The mutations are observed mostly in patients receiving at least 12 weeks of treatment. This maybe a bias which could explain the correlation with tumor volume: patients with longer exposure to treatment, who did not stop the treatment for toxicity before have by definition a higher likelihood to have a growing mass. This is difficult to assess in the absence of the full report of the clinical trial

Reviewer #5 (Remarks to the Author):

My opinion is that, in general, the referees had good points that were largely addressed by the authors. My opinions on this discussion are below:

Reviewer 1

1. R1 stated "The fact that resistance mutations can be found in ctDNA at low frequency has been demonstrated several times over now,"

the authors comments are valid;

A. Although resistance mutations in tumor treated with other targeted drugs have been described, emergence of TP53 mutations from patients treated with an MDM2 inhibitor has never been reported until now.

My opinion is that evidence, not expectations, are important.

2. R1 stated "despite the authors claims, a number of studies have now shown MDM2 inhibitors are ineffective with TP53 alterations"

and the authors comments are valid;

A. Although indeed it has been reported that MDM2 inhibitor is inactive in pre-clinical tumor models with TP53 mutation, it only raised a pre-clinical hypothesis that acquired mutation of TP53 is likely to be a major resistance mechanism to this class of drugs. Our study provides the first in vivo evidence supporting that hypothesis in cancer patients treated with an MDM2 inhibitor.

B.

There are several experimental compounds in this class in the early stages of clinical development and publication of our finding will be highly informative and could potentially reshape future clinical development strategies of this class of drugs.

My opinion is that this current manuscript provides evidence that is important to integrate into future studies in this class of drugs, which are now emerging, despite what predictions or expectations might have pre-existed. Evidence based science is often understated (in favour of

opinion) in choice of clinical cancer programmes.

3. R1: Perhaps most importantly, this is far too small a sample size to draw any definitive conclusions"

and the authors comments are valid;

A. Despite the small sample size, we demonstrate a robust and statistically significant finding that the total TP53 mutation burden and tumor growth are significantly correlated (Figure 2E).

My opinion is that this sample size is adequate to provide proof of concept in a rare disease. It is not trivial to acquire a large numbers of patients with this type of tumour for a clinical trial. Patients with such rare disease deserve evidence based clinical data to inform future choice of medicines; in this case, p53 reactivating drugs.

Reviewer 2 and 3

R2 and 3 had excellent points whose discussion in the manuscript would improve the quality of the manuscript. However, since this manuscript addressed the ever more common theme of emergent mutations; these common comments by reviewers 2 and 3 (about mechanism of acquired mutations) have not necessarily been adequately addressed to raise impact:

R2 2. "What is the evidence that this is TUMOR DNA vs normal DNA? and R3. "It would be very important to have documentation of the absence of mutation in genes unrelated to the MDM2/p53 pathway in cfDNA to confirm the hypothesis that this was a selection pressure."

If data from this manuscript are to, indeed, inform future clinical developments, then assays that measure mechanism of acquired resistance have to be incorporated (and explained in the discussion). For example, will a patients tumour have to undergo ultra deep DNA sequencing (to detect very rare p53 mutant clones) that were pre-existing (as in Optimizing Cancer Genome Sequencing and Analysis; DOI: <http://dx.doi.org/10.1016/j.cels.2015.08.015>). If pre-existing p53 mutations are detected and therefore mutant clones will evolve, is there any point in treating such patients? Alternatively, if ultra-deep sequencing essentially shows no evidence of pre-existing p53 mutant clones, would such patients be more likely to benefit in the medium term? Or, if p53 mutations in fact develop post-drug treatment, would this warrant combination trials of MDM2 drugs with DNA repair drugs to reduce extent of mutated clones being selected, etc? In other words, the mechanism of this acquired selection of mutant p53 clones to inform future clinical protocol development needs to be explicit stated to raise the impact for readers of this study.

Finally, these mdm2 amplified sarcomas form a typically rare disease. There are almost no solution options for patients with such rare diseases. Publishing such cumbersome (ethically constrained), clinically acquired data in a high profile journal, which provides incremental, if predictable evidence, is important. This could further drive discussions and debate that in turn improve the next iteration of clinical studies on mdm2 targeted drugs.

Author response to reviewer comments_2

Manuscript ID: NCOMMS-16-00144-T

Responses to reviewer #2

This interesting report describes the emergence of mutations in cfDNA in patients with liposarcoma with HDM2 amplification receiving HDM2 inhibitor SAR405838. The results of the clinical trial in terms of tumor control and side effects are not described, and will be described in a subsequent report. This is an important limitation of the study.

As indicated in the manuscript, the clinical results in terms of efficacy and toxicity have been previously reported (EORTC 2014, reference #8 in the manuscript). While a clinical manuscript based on the EORTC poster is in preparation, we do not feel that there is any additional clinical result not mentioned in our manuscript that alters the conclusions of our manuscript. We do describe the clinical findings that are important to support our conclusions. For example, we show the genetic status of patients at baseline, we show the kinetics of p53 mutation emergence, we show changes in tumor size by CT over time, we show that PK is not different between patients treated for more or less than 12 weeks, we show which patients had samples collected at which time points, we indicate when patients discontinued SAR405838 treatment, and we indicate that there were no objective responses. In addition, the clinical team felt that the demonstration of the emergence of p53 mutations in ctDNA in human patients was a technical and scientific finding that warranted a separate report, given the technical nature of assay development, the novelty of the result in human patients, and the potential for broad impact on the clinical development of MDM2 antagonists. As the additional clinical data do not alter the central conclusions of our manuscript, and a clinical manuscript based on the EORTC poster is in preparation, we do not feel it is feasible (or necessary) to add more clinical data to our manuscript.

Indeed, none of the patients have truly responded to the treatment, and therefore it remains speculative to mention primary or secondary resistance in tumors which are maybe only slowly progressive.

In the manuscript, we do not claim that these mutations are the direct cause of primary or secondary resistance in the clinical study. Indeed, as we indicated that there were no objective responses, this cannot be secondary resistance. We show that patients were p53 wild-type at baseline, p53 mutations rapidly emerged upon drug treatment, and the mutant allele burden correlates with tumor size. As SAR405838 activates wild-type p53, and pre-clinical data have demonstrated that the *de novo* acquisition of p53 mutations results in acquired resistance to HDM2 antagonists, we believe these data support the hypothesis that p53 mutation emergence, where no p53 mutation was detected at baseline, represents the growth of cells that would be resistant to drug treatment in response to selective pressure applied by the activation of p53. That is the extent of our claim in the manuscript, and we don't feel that any alteration to our manuscript can more adequately address this. However, we have amended the sentence in the discussion, from "Our results are consistent with several preclinical studies and provide the first *in vivo* evidence in support of the hypothesis that mutations in *TP53* is the major resistance mechanism to MDM2 inhibitors in the clinic" to "Our results are consistent with several

preclinical studies and provide the first in vivo evidence in support of the hypothesis that ***mutations in TP53 contribute to MDM2 inhibitor resistance*** in the clinic”

Another limitation of the study, pointed out by the authors is that rebiopsies were not performed. How can we be sure that mutations do not arise from other tissues?

As the reviewer mentions, we point this out in our manuscript. We agree that in order to be 100% sure that mutations do not arise from other tissues, re-biopsies would be required. However, making post-treatment biopsies mandatory, particularly biopsies upon progression, is not practically feasible. We did make post-treatment biopsies optional in the Phase 1 study, and encouraged their collection, but none were obtained. In fact, this is one of the major areas of added value for ctDNA – obviating the need for post-treatment tumor biopsies. In addition, in this indication (DDLPS), biopsies can have low cellularity, they frequently have a “mottled” appearance with areas of well- and de- differentiated cells, and frequently fail tissue requirements for sequencing. As such, while we agree that the ideal dataset would have included full, serial, post-treatment tumor biopsies (including upon progression), this was simply not possible within the ethical, financial, and practical constraints of a Phase 1 study in DDLPS. Unfortunately, there is nothing we can do in order to address this comment.

The correlation with tumor progression is only a correlative not demonstrative.

We do not claim causality. We simply show that changes in mutant allele burden correlates with changes in tumor size. We do believe that this result supports the hypothesis that the drug is applying selective pressure to the tumor and that p53 mutations arise from tumor cells. But we agree with this statement and we do not claim otherwise in the manuscript. We have added a sentence to discussion to explicitly say that this correlation does not necessarily indicate causality.

The method used to assess the presence/emergence of mutated p53 are appropriate. Most mutations are deleterious according to the public databases. The mutations are observed mostly in patients receiving at least 12 weeks of treatment. This maybe a bias which could explain the correlation with tumor volume: patients with longer exposure to treatment, who did not stop the treatment for toxicity before have by definition a higher likelihood to have a growing mass. This is difficult to assess in the absence of the full report of the clinical trial.

We agree that this finding indicates that patients who are treated for longer periods of time are have more time to accumulate mutations. We do address this in the manuscript by saying, “Therefore, it is possible that the patients treated for less than 12 weeks were not exposed to SAR405838 long enough to develop *TP53* mutations, and had they been treated long enough, *TP53* mutations may have emerged.” We don’t believe the increase in p53 mutation burden can simply be a marker of tumor burden. Patients presented with detectable tumors which in some cases were quite large before the initiation of treatment, yet had no mutations. What we saw was that the change in p53 mutation burden correlated with change in tumor size once treatment was initiated. We agree that patients with a higher likelihood of a growing mass on treatment also have increasing p53 mutation allele burden, which is consistent with what we say in the manuscript. Our claim is only that change in mutation

burden correlated with change in tumor size on treatment. As indicated above, we don't claim causality. As such, we don't believe there is any data in the full clinical report that would change this interpretation. However as stated above, we have added a statement to the discussion section of the manuscript explicitly stating that this correlation does not necessarily indicate causality.

Response to reviewer #5

If data from this manuscript are to, indeed, inform future clinical developments, then assays that measure mechanism of acquired resistance have to be incorporated (and explained in the discussion). For example, will a patient's tumour have to undergo ultra deep DNA sequencing (to detect very rare p53 mutant clones) that were pre-existing (as in *Optimizing Cancer Genome Sequencing and Analysis*; DOI: <http://dx.doi.org/10.1016/j.cels.2015.08.015>). If pre-existing p53 mutations are detected and therefore mutant clones will evolve, is there any point in treating such patients? Alternatively, if ultra-deep sequencing essentially shows no evidence of pre-existing p53 mutant clones, would such patients be more likely to benefit in the medium term? Or, if p53 mutations in fact develop post-drug treatment, would this warrant combination trials of MDM2 drugs with DNA repair drugs to reduce extent of mutated clones being selected, etc? In other words, the mechanism of this acquired selection of mutant p53 clones to inform future clinical protocol development needs to be explicit stated to raise the impact for readers of this study.

Given space and word count constraints when preparing this manuscript, a more extensive discussion of how p53 mutations could be measured and what action could be taken was not included. However, we do include the key implication, which is: "Given the rapidity with which TP53 mutations emerged, we hypothesize that HDM2 antagonists will likely require development in combination with other agents. Various combination partners, including a mutant p53 activating drug like APR-26, MEK or PI3K inhibitors, or Bcl-2 inhibitors, have been proposed. Ongoing clinical studies may help determine if combinations can improve the clinical activity of HDM2 antagonists."

In other words, the main implication is that HDM2 antagonists will have to be developed in combination, including possibly drugs that activate mutant p53 like APR-246. We believe our data show that it is not necessary to have a patient's tumor undergo ultra deep sequencing, as ctDNA sequencing can clearly detect p53 mutations, and mutations may arise *de novo*. Our data also show that p53 mutations will develop post-treatment, and therefore combinations will be required. Again, given constraints, that is what we focus on in the discussion. As we show that these mutations may not be pre-existing subclones, even if a patient's tumor showed no mutations after ultra-deep sequencing at baseline, we believe that mutations could arise *de novo*, and therefore these patients may not be more likely to benefit.

We would be happy, if allowed by constraints, to expand this discussion to include more of the above points. While we feel that our discussion does address this main conclusion, we are happy to expand it if that is desired.